# The Antitumor Effect of the DNA Polymerase Alpha Inhibitor ST1926 in Glioblastoma: A Proteomics Approach

**DOI:** 10.3390/ijms241814069

**Published:** 2023-09-14

**Authors:** Chirine El-Baba, Zeinab Ayache, Mona Goli, Berthe Hayar, Zeinab Kawtharani, Claudio Pisano, Firas Kobeissy, Yehia Mechref, Nadine Darwiche

**Affiliations:** 1Department of Biochemistry and Molecular Genetics, American University of Beirut, Beirut 1107 2020, Lebanon; chirinebaba@gmail.com (C.E.-B.); zja09@mail.aub.edu (Z.A.); bh48@aub.edu.lb (B.H.); zmk30@mail.aub.edu (Z.K.); fkobaissy@msm.edu (F.K.); 2Department of Chemistry and Biochemistry, Texas Tech University, Lubbock, TX 79409, USA; mona.goli@ttu.edu; 3Biogem, Institute of Molecular Biology and Genetics, 83031 Ariano Irpino, Italy; claudio.pisano@biogem.it; 4Department of Neurobiology, Center for Neurotrauma, Multiomics and Biomarkers (CNMB), Morehouse School of Medicine, 720 Westview Dr. SW, Atlanta, GA 30310, USA

**Keywords:** glioblastoma, ST1926, POLA1 inhibitor, proteomics, biomarkers

## Abstract

Glioblastoma Multiforme (GBM) is the most aggressive form of malignant brain tumor. The median survival rate does not exceed two years, indicating an imminent need to develop novel therapies. The atypical adamantyl retinoid ST1926 induces apoptosis and growth inhibition in different cancer types. We have shown that ST1926 is an inhibitor of the catalytic subunit of DNA polymerase alpha (POLA1), which is involved in initiating DNA synthesis in eukaryotic cells. POLA1 levels are elevated in GBM versus normal brain tissues. Therefore, we studied the antitumor effects of ST1926 in several human GBM cell lines. We further explored the global protein expression profiles in GBM cell lines using liquid chromatography coupled with tandem mass spectrometry to identify new targets of ST1926. Low sub-micromolar concentrations of ST1926 potently decreased cell viability, induced cell damage and apoptosis, and reduced POLA1 protein levels in GBM cells. The proteomics profiles revealed 197 proteins significantly differentially altered upon ST1926 treatment of GBM cells involved in various cellular processes. We explored the differential gene and protein expression of significantly altered proteins in GBM compared to normal brain tissues.

## 1. Introduction

Glioblastoma Multiforme (GBM) is the most aggressive primary brain tumor in adults and is considered the deadliest type of all gliomas. GBM accounts for 54% of all gliomas, 45% among primary malignant brain and central nervous system tumors, and 16% among all primary brain tumors [1]. The current treatment against glioblastoma is a multimodal regimen combining maximal safe surgery resection, adjuvant radiotherapy, and chemotherapy using temozolomide, a small-molecule alkylating drug [2]. Despite major improvements in GBM management, GBM has one of the poorest survival rates of all malignant brain tumors. GBM patients have a median survival of 15 months, and their survival after diagnosis and treatment ranges between one and five years [3]. Unfortunately, resection does not fully isolate the tumor because of its infiltrative nature [4], and traditional cytotoxic chemotherapeutics are often unsuccessful due to the high tumor heterogeneity and drug resistance. The toxicity of chemotherapeutic agents and the complexity of GBM underscore the need for more efficient and targeted treatments for this type of cancer. Moreover, GBM has an immunosuppressive tumor microenvironment that exhibits tumorigenic and immunosuppressive phenotypes through tumor-associated macrophages. GBMs have a pronounced heterogeneity in oncogenic de novo and recurrent mutations. Therefore, there is a major need to develop specific therapies for recurrent GBM [5].

Retinoids are promising drugs for treating several cancers, and studies have shown their potential for glioblastoma treatment [6]. Natural retinoids have some limitations in cancer therapy due to their toxicity and acquired drug resistance. Consequently, synthetic retinoids were developed with improved selectivity and reduced toxicity [7]. We and others have reported that the synthetic adamantyl retinoid ST1926 is a potent and selective anticancer drug in various preclinical tumor models, such as the human ovarian tumor, neuroblastoma, and rhabdomyosarcoma [8,9,10]. Furthermore, ST1926 reduces prostate cancer cell growth, migration, and invasion; induces p53-independent apoptosis via early DNA damage; and inhibits tumor growth in mouse prostate cancer xenografts [11]. ST1926 was also shown to inhibit the proliferation and induce cell death in human colorectal cancer (CRC) cell lines independent of p53 and p21 status. We have demonstrated that ST1926 is a potent DNA polymerase alpha (POLA1) inhibitor [12]. Recently, ST1926 was shown to induce apoptosis and autophagy, impair mitochondrial function in glioma cells by controlling complex II, and suppress tumor growth in a glioma xenograft mouse model [13].

Interestingly, we found POLA1 expression levels to be elevated in human GBM tissues compared to their normal counterparts. Therefore, we studied and characterized the antitumor effects of ST1926 in human GBM cell lines. Using a proteomics approach, we surveyed the global protein profile in human GBM cell lines upon treatment with ST1926 using liquid chromatography coupled with tandem mass spectrometry (LC-MS/MS) to identify new molecular targets of ST1926. These identified biomarkers might reveal high-potential targets to be considered in future strategies for treating GBM.

## 2. Results

### 2.1. POLA1 Levels Are Elevated in GBM Tissues and Are Reduced in GBM Cells upon ST1926 Treatment

We determined POLA1 levels in GBM using in silico analysis. The Oncomine database identified three different studies comparing POLA1 in GBM patient tissues versus normal brain counterparts [14,15,16]. These studies revealed that GBM tissues have significantly higher POLA1 expression levels than normal brain tissues (Figure 1A).

ST1926 was shown to mediate its antitumor activities through DNA damage induction, mainly by inhibiting POLA1 activity in cancer cells. Since POLA1 levels are elevated in GBM tissues, we evaluated the effect of ST1926 on POLA1 protein levels in GBM cells. The human GBM cell lines U251 and U87MG cells showed decreased levels of POLA1 as early as 6 h post ST1926 treatment (Figure 1B).

### 2.2. ST1926 Inhibits Cell Growth of GBM Cell Lines

We investigated the effect of ST1926 on the growth of different GBM cell lines using the 3-(4,5-dimethylthiazol-2-yl)-2,5-diphenyl-2H-tetrazolium bromide (MTT) assay. We treated the human GBM cell lines U251, U87MG, A172, and U118 with different concentrations of ST1926. Micromolar (µM) concentrations of ST1926 significantly inhibited cell growth in a time- and dose-dependent manner in all tested GBM cell lines (Figure 2). To calculate the exact IC_50_ concentrations of the different cell lines, we had to use high µM concentrations for U87MG, A172, and U118. The IC_50_ for ST1926 treatment at 72 h was 0.1, 0.4, 7.5, and 10 μM in U251, U87MG, A172, and U118 cells, respectively. We decided to study the mechanism of action of ST1926 in U251 and U87MG since these two cell lines were the most sensitive to the compound. The effect of ST1926 on cell growth was further confirmed in the U251 and U87MG cells by the sulforhodamine B (SRB) assay for cell density determination, based on the measurement of cellular protein content, an indirect measure of cell growth. We observed similar trends upon ST1926 treatment by MTT and SRB assays (Appendix A).

### 2.3. ST1926 Treatment of GBM Cells Results in G_0_/G_1_ Cell Cycle Arrest, Apoptosis, and DNA Damage

Cell cycle analysis was performed using flow cytometric analysis of DNA content to determine the mechanism of action of ST1926 in GBM-induced cell growth inhibition. U251 and U87MG cells were treated with 0.5 μM ST1926 for three days and stained with propidium iodide. ST1926 resulted in cell accumulation in the sub-G1 phase in both cell lines (Figure 3). The percentage of U251 cells in the sub-G_1_ phase increased from 4% in the control to 17% after three days of ST1926 treatment (Figure 3A) and from 2% in the control to 40% in the ST1926-treated U87MG cells (Figure 3B). G_0_/G_1_ cell cycle arrest was observed in ST1926-treated U251 and U87MG cells, where it reached 80% of the treated cells at 24 h (Figure 3A,B). We observed a significant decrease in the S phase of U87MG-treated cells. BrDu staining would confirm the effect of ST1926 treatment on the S-phase distribution of the cells.

To confirm apoptosis induction by ST1926 treatment, we performed a terminal deoxynucleotidyl transferase dUTP nick and labeling (TUNEL) assay to detect double-stranded DNA cleavage, a late apoptotic event. U251 and U87MG cells were treated with 0.5 μM of ST1926. The percentage of TUNEL-positive cells increased from 6% in the control to 80% in the U251-treated cells (Figure 3A) and from 6% in the control to 38% in the U87MG-treated cells (Figure 3B). In addition, PARP was cleaved as early as 24 h in ST1926-treated U251 cells and at 48 h in the U87MG-treated cells (Figure 3A,B). We also determined the effect of ST1926 on DNA damage in GBM cells. We observed elevated levels of γ-H2AX, a DNA damage marker, in a time-dependent manner, starting as early as 6 h post treatment in tested GBM cell lines (Figure 3A,B).

### 2.4. Proteomics Analysis of ST1926-Treated GBM Cells by LC/MS-MS

We treated GBM cell lines with ST1926 for different time points for LC-MS/MS-based proteomics analysis in three independent experiments. U251 cells were treated with 0.5 µM ST1926 for 2 and 24 h, while U87MG and U118 cells were treated with 0.5 µM ST1926 for 2, 24, and 48 h. Control samples were collected at the maximal hour of treatment. In the following sections, we show the protein expression of different GBM cell lines after treatment with ST1926 at different time points by employing Principal Component Analysis (PCA), Volcano plots, and Hierarchical Heatmap Clustering.

#### 2.4.1. Protein Expression in GBM Cell Lines after Treating with ST1926 Using Principal Component Analysis

Principal Component Analysis (PCA) plots illustrate the transformation of the comprehensive changes of protein levels in large datasets into two-dimensional plots to illustrate the alterations amongst the dissimilar groups. The PCA plots (Figure 4) depict the PCA of the different conditions of ST1926 treatment in the U251 and U87GM human GBM cells, including their triplicates. The PCA plots illustrate a significant separation among the various groups, the control, and the treated samples at different time points, thus signifying that ST1926 generated distinctive proteomics profiles in treated samples compared to control counterparts. The triplicates are clustered together, suggesting the excellent reproducibility of the LC-MS/MS-based proteomics analysis in this study. PCA results are provided for the ST1926-treated U118 cells in Appendix A.

#### 2.4.2. Protein Expression in GBM Cell Lines after Treating with ST1926 Using Volcano Plots

A volcano plot is a type of scatterplot used in proteomics research to identify differentially expressed proteins. The plot displays the log twofold change on the x-axis and the negative log10 *p*-value on the y-axis. The significantly up-regulated or down-regulated proteins are represented by proteins far away from the center of the plot, resembling a volcano. The volcano plots below visualize the statistical protein expression significance of the different conditions of ST1926 treatment in the U251, U87MG, and U118 human GBM cells (Figure 5). Proteins with statistically significant differences (*p*-value < 0.05) between different groups were selected using a *t*-test (Table 1). The list of up- and down-regulated proteins in each comparison is provided in Appendix A, along with their accession numbers, gene names, and fold changes.

#### 2.4.3. Protein Expression in GBM Cell Lines after Treating with ST1926 Using Hierarchical Heatmap Clustering

Differences in significant protein expressions between groups were also visualized using hierarchical heatmap clustering. In the heatmaps, each row represents the abundance of a significant protein in two compared groups, while each column is the replicate of each group. The green color denotes a low relative abundance, while red indicates a high relative abundance. Figure 6 represents the expression of the proteins with significant changes in the U251 cell line at different time points, and Appendix A represent the expression of the proteins with significant changes in U118 and U87MG cell lines using heatmaps.

#### 2.4.4. Comparative Protein Expression upon ST1926 Treatment of GBM Cell Lines

To further analyze the expression of proteins upon ST1926 treatment, we constructed a Venn diagram to investigate the comparative expression among the three human GBM cell lines after 24 h of ST1926 treatment (Appendix A). Regarding the mass spectrometry analysis, 148 hits (proteins) were unique to the U251 cells, 51 hits were unique to the U118 cells, and 114 hits were unique to the U87MG cells upon 24 h of treatment of ST1926. In total, 39 hits were common between the U251 and the U87MG cells upon ST1926 treatment for 24 h; U251 shared 6 hits with U118 cells; and U87MG shared 10 hits with U118 cells (Table 2). Four proteins were found to be regulated in the same manner in the three ST1926-treated GBM cell lines: DNA topoisomerase 2-alpha (TOP2A), Sequestosome-1/p62 (SQSTM1), Tubulin alpha-4A chain (TUBA4A), and Collagen alpha-1(VI) chain (COL6A1). Similarly, these proteins remained up- or down-regulated in U87MG and U118 cell lines after 48 h of ST1926 treatment (Figure 7).

### 2.5. Protein Pathway Analysis of the Differentially Expressed Proteins in ST1926-Treated GBM Cells

Following the mass spectrometry analysis, we constructed protein–protein interaction pathways associated with the differentially expressed proteins in ST1926-treated GBM cells using Pathway Studio analysis version10.001. This software allows researchers to explore the interactions among molecules, cell processes, and diseases and draw them out using literature-based evidence. We selected to study the protein pathway analysis in the U251 cells, as these were the most sensitive to ST1926 treatment. The pathways demonstrate the link between up- or down-regulated proteins in U251 cells upon ST1926 treatment for 24 h (Figure 8A–C). There were three identified group links: Group A links expressed proteins to cell proliferation, cell growth, cell cycle, and cell cycle arrest; Group B links expressed proteins to DNA replication, DNA repair, DNA damage, response to DNA damage, and oxidative stress; and Group C links expressed proteins to cell survival, apoptosis, and cell death. Similarly, we applied the same altered differential protein hits to study gene overlap (Appendix A), gene ontology clustering (Appendix A), and protein interaction-network annotation function (Appendix A) via MetaScape Software (https://metascape.org/). Data from MetaScape analysis corroborated Pathways Studio analysis, where we identified the top enriched pathway as the Cell cycle pathway, RNA Metabolism, and other signaling pathways. In addition, protein–protein interaction–network annotation function analysis identified RNA interactions/assembly and metabolic pathways to be among the top hits.

### 2.6. Relevance of Differentially Altered Genes by ST1926 Treatment between Normal Brain and Glioblastoma Tissues

We studied the relevance of the significantly altered genes by ST1926 treatment in U251 cells in GBM tumorigenesis. We checked significantly up- or down-regulated proteins after 2 and 24 h of treatment of U251 cells with 0.5 μM ST1926. In total, 50 proteins were obtained. We sought to study whether these proteins and their corresponding transcripts were associated with tumorigenesis in GBM. We investigated the differential gene expression by evaluating mRNA levels between normal brain tissues and those of glioblastoma presented in a boxplot, using Gene Expression Profiling Interactive Analysis (GEPIA). For the same genes, we explored the protein levels using The Human Protein Atlas (https://www.proteinatlas.org/ accessed on 2 May 2023) in normal brain tissues (cerebral cortex) compared to glioblastoma tissue immunohistochemistry (IHC) using tissue and pathology atlases.

We first focused our analysis on transcripts and proteins that were up-regulated in GBM tumorigenesis (Figure 9). The solute carrier family 2 facilitated glucose transporter member 1 (SLC2A1), and the solute carrier family 25 members (SLC25A6) transcripts and proteins were up-regulated in glioma versus normal brain tissues (Figure 9A,B). Furthermore, the chromobox protein homolog 3 (CBX3), DEK proto-oncogene (DEK), and DEAD (Asp-Glu-Ala-Asp) box polypeptide 39B (DDX39B) transcripts and proteins were up-regulated in glioma versus normal brain tissues (Figure 9C–E). Interestingly, ST1926 treatment of GBM cells significantly down-regulated these proteins.

Next, we studied transcripts and proteins down-regulated in GBM tumorigenesis (Figure 10). Phosphatidylethanolamine-binding protein 1 (PEBP1) and Hsp70-binding protein 1 (HSPBP1) were reduced in glioma versus normal brain tissues (Figure 10A,B). Of interest, ST1926 treatment of GBM cells significantly increased these proteins. Peptidyl arginine deiminase, type II (PADI2), and UMP-CMP kinase (CMPK1) transcripts were similar between glioblastoma and normal brain tissues. However, proteins were down-regulated in glioblastoma versus normal brain tissues (Figure 10C,D).

These findings highlight the potential of ST1926 in reducing the levels of relevant oncoproteins and increasing tumor-suppressing proteins in GBM tumorigenesis.

## 3. Discussion

GBM is a grade IV glioma and is the most common and deadliest primary brain tumor. This highly invasive neoplasm remains incurable, despite relentless efforts to improve and increase the long-term benefits of GBM therapy. Advancements toward curing and enhancing the long-term survival of GBM patients are limited by several challenges. GBM tumors cannot be totally removed due to their rapid cell proliferation and extensive invasion into the surrounding brain tissues [17]. Moreover, immunosuppression, tumor heterogeneity, the restrictive nature of the blood–brain barrier, and chemo-resistant pathways limit the effectiveness of drugs, including temozolomide, the treatment of choice against GBM [18]. Thus, there is an imminent need to develop safer and more effective therapeutic strategies.

Retinoids are used as chemotherapeutic and chemopreventive agents in cancer management [19]. However, natural retinoids have limited antitumor abilities and are often associated with undesirable side effects and acquired drug resistance [20]. Their synthetic counterparts were developed with enhanced specificity and fewer side effects [7]. Synthetic retinoids, particularly those with an adamantyl moiety, displayed promising antitumor activities in various human tumor models [21]. ST1926, one of the most interesting compounds of these derivatives, displayed a favorable pharmacokinetic profile compared to its parental CD437 compound [22]. It was shown to inhibit tumor growth in several in vitro and in vivo cancer models, independent of p53 and retinoic acid receptor signaling pathways [23] in ovarian carcinoma; adult T-cell leukemia/lymphoma; rhabdomyosarcoma; and breast, CRC, and prostate cancer [8,11,12,24,25,26]. We have shown that ST1926 targets POLA1 and suppresses tumor growth in CRC models [12].

We found POLA1 expression levels to be elevated in human GBM tissues compared to their normal brain counterparts. Therefore, we studied and characterized the antitumor effects of ST1926 in human GBM cell lines. According to The Cancer Genome Atlas (TCGA) (TCGA 2013), the p53 signaling pathway is deregulated in 94% of GBM cell lines and 84% of GBM patients [27,28]. We have selected several GBM cell lines with a different p53 status, namely, U251 and A172 cells that have a mutated p53 gene, while U87MG and U118 have a wild-type counterpart [29]. Tumor cells with mutated p53 tend to resist radiation therapy, and patients with such tumors relapse in a short period of time [30]. In our study, ST1926 induced cell death independently of p53 status, which was also demonstrated in CRC [12], rhabdomyosarcoma [8], ovarian carcinoma [31], and breast cancer [24].

It was recently shown that ST1926 inhibits glioma progression and impairs mitochondrial complex II [13]. Importantly, ST1926 at concentrations as high as 10 μM displayed no significant cytotoxicity, nor any change in the morphology or induced excessive ROS production in normal human astrocytes [13]. We have observed that ST1926 is a powerful inducer of G_0_/G_1_ cell cycle arrest, apoptosis, and DNA damage, irrespective of p53 status.

Next, we examined the effect of ST1926 on POLA1 protein levels since ST1926 was shown to exert its anti-tumor effect through POLA1 inhibition [12]. POLA1 is the initiator of eukaryotic DNA replication [32]. The parental molecule of ST1926, CD437, was shown to target POLA1 and, thus, prevent cellular proliferation [33]. Similarly, we have shown that ST1926 reduces POLA1 protein levels and inhibits POLA1 activity in a concentration-dependent manner [12]. Since POLA1 is overexpressed in GBM, it is a potential target for ST1926.

Deciphering the molecular networks and pathways affected by ST1926 treatment in GBM cells is crucial to better understanding its mechanisms of action. The advent of proteomics analysis has revolutionized how we understand the cellular, molecular, and signaling processes of treated cells by a particular drug in pathology. The field of proteomics has gained significant importance, particularly in cancer studies, as it has contributed to the identification of biomarkers and revealed altered protein expression patterns that can be useful for tumor classification, prognosis, and treatment in a personalized fashion. Our LC-MS/MS analysis identified 197, 71, and 167 proteins in ST1926-treated U251, U118, and U87, respectively. In this study, we focused on U251 cells, which are sensitive to ST1926 at sub-µM concentrations and have a mutant p53. 

Following the LC-MS/MS proteomics analysis, we generated a protein pathway analysis linking these 197 altered proteins in ST1926-treated U251 cells to different biological processes and molecular functions using Pathway Studio software version 10.001, along with MetaScape, to interrogate gene overlap among the treatment and controls to evaluate gene ontology clustering and, finally, to assess protein interaction–network annotation function. Out of these 197 altered proteins, we selected the proteins that are of most interest in cancer research and therapy. We investigated in silico, using GEPIA and The Human Protein Atlas, the differential gene expression and protein expression of the selected proteins between GBM and normal brain tissues. Many of these proteins have been extensively studied in cancer research and therapy; therefore, it is essential to investigate how ST1926 regulates them. This will offer new insights into specific aspects and targets of ST1926 activity, which can be translated into novel and efficient therapeutic opportunities for patients with GBM.

In silico analysis revealed that the solute carrier family 2, facilitated glucose transporter member 1 (SLC2A1), also known as GLUT1 (Glucose transporter 1), is elevated in GBM versus normal brain tissues. The mRNA and protein levels of SLC2A1 (GLUT1) are higher in GBM than in the normal brain tissues, as indicated by GEPIA and The Human Protein Atlas immunohistochemical analyses, respectively. GLUT1, a primary mediator of glycolysis, is overexpressed in human GBM, with the highest expression levels in grade IV GBM compared to low-grade gliomas [34]. The patient population with high GLUT1 demonstrated worse survivorship, indicating a negative correlation between GLUT1 and patient prognosis. Silencing GLUT1, via its associated binding partner Tubulin 4, decreased cell viability, sphere formation ability, and cell aggressiveness, suggesting both as potentially druggable targets in GBM [34]. Another solute carrier, SLC25A6, belonging to the mitochondrial carrier family (MCF), is responsible for controlling the transfer of metabolites to the mitochondria through the inner mitochondrial membrane (IMM). Rochette et al. reviewed the role of MCF members, including SLC25A6, in cellular homeostasis and examined how their inhibition may lead to reprogramming cellular metabolism, compromising cell growth, and increasing cellular sensitivity to anticancer drugs [35]. Additionally, they compared the mitochondria of cancer to that of normal cells, showing a significant increase in IMM potentials and the number of transporters, including SLC25A5, which shares many features with SLC25A6 [34], to be strongly overexpressed in different types of cancers [36]. The gene expression analysis and IHC displayed overexpression of SLC25A6 in GBM versus normal brain tissues, suggesting that SLC25A6 may be an antitumoral target in GBM. Interestingly, our results indicate a down-regulation of both solute carrier proteins SLC2A1 and SLC25A6 in U251 cells post ST1926 treatment.

The chromobox protein homolog 3 (CBX3) is another protein of interest modulated by ST1926 in GBM-treated cells. CBX3 is a member of the heterochromatin protein 1 family. Its expression is elevated in osteosarcoma tissues and is associated with poor prognosis in osteosarcoma patients [37]. Recently, the oncogenic role of CBX3 has been identified in various malignant tumors and serves as a prognostic and immunological biomarker [38]. We have observed significantly higher transcripts and protein levels in GBM than in normal brain tissues. This is in accordance with a study by Zhao et al. that shows overexpression of CBX3 in glioma and how CBX3 knockdown in U87MG induces apoptosis and suppresses tumor formation [39]. Interestingly, our results demonstrate that ST1926 down-regulated CBX3 protein levels significantly.

The DEK proto-oncogene is preferentially overexpressed in actively proliferating and malignant cells [40]. We observed significant increases in DEK mRNA and protein levels in GBM versus normal brain tissues. DEK silencing in U251 cells induced apoptosis and cell cycle arrest, suggesting DEK’s role in promoting tumor formation and progression [41]. Another oncogene is DDX39B (DEAD (Asp-Glu-Ala-Asp) box polypeptide 39B), a member of the DEAD box (DDX) RNA helicase family [42]. We showed that DDX39B mRNA and protein levels are markedly elevated in GBM compared to normal brain tissues. A recent study by He et al. elucidated the role of DDX39B in facilitating the migration, invasion, and epithelial–mesenchymal transition (EMT) capacities of CRC cells and validated these findings in vivo [43]. Interestingly, ST1926 treatment targeted both the oncogenes DEK and DDX39B and down-regulated their corresponding expression in U251 cells.

Phosphatidylethanolamine binding protein 1 (PEBP1) is a member of the phosphatidylethanolamine-binding family of proteins and is a crucial modulator of several signaling pathways. It is widely recognized that PEBP1 inhibits the metastatic dissemination of tumor cells. The down-regulated expression of PEBP1 is observed in several human cancers, which has defined it as a metastasis suppressor gene. In hepatocellular carcinoma, the down-regulation of PEBP1 resulted in aggressive tumor behavior and poor prognosis [44]. Our in silico analysis showed PEBP1 down-regulation in GBM at the mRNA and protein levels compared to normal brain tissues. Decreased levels of PEBP1 were observed in high-grade gliomas compared to the non-neoplastic tissues and lower-grade gliomas [45]. Interestingly, ST1926 up-regulated this metastasis suppressor protein in GBM cells.

Peptidyl arginine deiminase, type II (PADI2), belongs to the peptidyl arginine deiminase family of enzymes and has been implicated in human neurodegenerative disorders. While PADI2 is overexpressed in various cancer types, and its overexpression is correlated with worse clinical outcomes, it is down-regulated in CRC, which correlates with poor prognosis [46]. Gene expression analysis showed a non-significant increase in PADI2 mRNA levels in GBM versus normal brain tissues; however, IHC analysis showed lower protein levels in glioma than in normal brain tissues. ST1926 up-regulated PADI2 protein levels in GBM cells.

Cytidine monophosphate (UMP-CMP) kinase 1 (CMPK1) is responsible for the metabolism of CMP, UMP, and deoxycytidine analogs, many of which are considered anticancer agents. Recent findings have shown that elevated CMPK1 expression correlates with extended survival and enhanced responsiveness to 5-FU therapy in gastric cancer. Conversely, the suppression of CMPK1 resulted in the subjection of gastric cancer cells to DNA damage and cell death following 5-FU treatment [47]. CMPK1 is also of predictive and prognostic value in several cancers [48]. The role of CMPK1 is not yet elucidated in GBM. The IHC analysis showed lower CMPK1 levels in GBM versus normal brain tissues; ST1926 up-regulated CMPK1 levels in the GBM cell line.

Heat shock protein 70-binding protein 1 (HspBP1), an Hsp70 co-chaperone, is highly expressed in several cancers, including breast cancer, and inversely correlates with tumor aggressiveness [49]. HspBP1 regulates the phosphorylation of the tumor suppressor BRCA1 and its subsequent recruitment to DNA damage sites. It was recently shown that HspBP1 inhibits, in a BRCA1-dependent manner, the tumorigenesis of breast cancer in vitro and in vivo and promotes apoptosis [50]. Although we did not observe any clear change in HspBP1 levels between GBM and normal brain tissues, our LC-MS/MS analysis demonstrated ST1926 up-regulated HspBP1 levels in U251 cells. This is in accordance with the study by Tanimura et al. showing that anticancer drugs increase the levels of HspBP1 expression, which subsequently forms a specific interaction with Hsp70, counteracting its ability to prevent the cell death-associated permeabilization of lysosomes. Consequently, HspBP1 plays a role in promoting programmed cell death through cathepsin activation in response to anticancer drugs in tumor cells [51].

In summary, the proteomics profiles highlighted the favorable signaling pathways regulated by ST1926 in GBM cells. ST1926 treatment down-regulated a group of oncogenic proteins (SLC2A1, SLC25A6, CBX3, DEK, DDX39B) and up-regulated a group of presumably tumor-suppressing proteins (PEBP1, HspBP1); PADI2 and CMPK1, of unknown function in GBM, were also up-regulated. The evidence that some proteins affected by ST1926 are also mis-regulated in tumors teaches us crucial facts about glioblastoma biology and therapeutic strategies. These identified novel proteins may be potential therapeutic targets for new drug development in glioblastomas, especially using POLA1 inhibitors. These proteins are implicated in several aspects of crucial cellular processes, such as cell proliferation, apoptosis, cell death, invasion, metastasis, and metabolism. They may be studied as potential therapeutic targets for ST1926, which will pave the way for patient-specific targeted therapies.

## 4. Materials and Methods

### 4.1. Cell Culture

U87MG, U251, A172, and U118 human GBM cell lines (American Tissue Culture Collection, ATCC) were cultured at 37 °C in a humidified incubator (95% air, 5% CO_2_) and grown in Dulbecco’s Modified Eagle’s medium/nutrient mixture F-12 Ham (DMEM-F12) (Sigma Aldrich, St. Louis, MO, USA), supplemented with 10% fetal bovine serum (FBS) (Sigma Aldrich, St. Louis, MO, USA), 1% of penicillin-streptomycin (Sigma Aldrich, St. Louis, MO, USA), 1% non-essential amino acids (Sigma Aldrich, St. Louis, MO, USA), and 1% sodium pyruvate (Sigma Aldrich, St. Louis, MO, USA).

### 4.2. Drug Preparation

ST1926 was obtained from Biogem Institute (Ariano Irpino, Italy) and was reconstituted in 0.1% dimethylsulfoxide (DMSO) at a concentration of 0.5 M, aliquoted, then stored at −80 °C. First, 0.5 M ST1926 aliquots were diluted in pure ethanol to reach 10^−2^ M, then were serially diluted with fresh media for use at 0.5 μM.

### 4.3. Cell Viability MTT Assay

3-(4,5-dimethylthiazol-2-yl)-2,5- diphenyltetrazolium bromide (MTT) was used to assess cell viability. U87MG, U251, A172, and U118 cells were seeded in triplicates in 96-well plates at a density of 5000 cells/well. Cells were treated with different concentrations of ST1926, ranging from 0.01 μM to 100 μM, for up to three days. For each time point, 10 mg/mL thiazolyl blue tetrazolium bromide dye (MTT, Sigma Aldrich) was added to each well (final concentration 1 mg/mL). The formed intracellular formazan crystals were dissolved by adding 100 μL of SDS-based solubilizing agent, and values were measured at a wavelength of 595 nm using an ELISA microplate reader (Multiskan Ex). The cell viability was expressed as percentage growth relative to control untreated cells.

### 4.4. Cell Cycle Analysis

U87MG and U251 cells were seeded into 100 mm dishes at a density of 1 × 10^6^ cells/dish and treated with 0.5 μM of ST1926 for up to three days. Then, cells were fixed with 80% cold ethanol. The cell pellets were treated for 45 min with 100 μL of 200 μg/mL DNase-free RNase A. Cells were stained with 15 μL of 1 mg/mL propidium iodide (PI) (Sigma Aldrich, St. Louis, MO, USA). The cellular DNA content and PI fluorescence were measured using flow cytometry (FACScan, Becton Dickinson, Franklin Lakes, NJ, USA). A total of 10,000 gated events were acquired to assess the proportions of cells at different stages of the cell cycle.

### 4.5. TUNEL Assay

The terminal deoxynucleotidyl transferase dUTP nick end labeling (TUNEL, Roche Diagnostics, Indianapolis, IN, USA) assay was used to measure apoptosis. Cells were seeded at a density of 1 × 10^6^ cells/ flask in 75 cm^2^ flasks, then treated with 0.5 µM ST1926, or left untreated for the controls. Two controls, positive and negative, were prepared for use in the experiment. At the indicated time point, cells were collected by trypsinization, washed with 1% BSA in 1X PBS, and then fixed with 4% formaldehyde for 30 min at room temperature. The next steps were followed according to the manufacturer’s instructions. A total of 10,000 gated events were acquired to assess the proportions of apoptotic cells quantified with the excitation wavelength set at 470–490 nm and the emission wavelength at 505 nm.

### 4.6. Cell Collections for Proteins Profiling

U251, U87MG, and U118 cells were seeded in triplicates in three independent experiments into 100 mm dishes at 1 × 10^6^ cells/dish density. When cell confluency reached 40–50%, cells were treated with 0.5 μM of ST1926 for up to 48 h. A control dish was used for each experiment in which cells were only treated with a solvent medium for up to 48 h. At each time point, cells were collected and then lysed using Nonidet™ P 40-based lysis buffer 1% (*v*/*v*).

### 4.7. Proteomics Analysis

Proteomics analysis was performed to identify candidate proteins with statistically significant differences between control groups (solvent-treated) and the cell lines treated with 0.5 µM ST1926 at different time points. A total of 33 cell line samples representing three independent experiments were conducted on three cell lines (U251 [9 samples], U118 [12 samples], and U87MG [12 samples]) using 0.5 µM ST1926, then shipped on dry ice to the Proteomics Core Facility (Mechref Omics Lab at the Chemistry and Biochemistry Department, Texas Tech University, Lubbock, TX, USA). U87MG and U118 cells were treated with 0.5 µM ST1926 for 2, 24, and 48 h, while U251 cells were treated with 0.5 µM ST1926 for 2 and 24 h (control collected at the maximal hour of treatment). Note that the maximal hour of treatment was chosen according to the time needed for 0.5 μM ST1926 to inhibit 50% of cell growth.

#### 4.7.1. Cell Lysis and Protein Extraction

Frozen cells were thawed at room temperature. Then, they were mixed with 5% sodium deoxycholate (SDC) solution with 400 µm molecular biology-grade zirconium beads (BMBZ 400-250-36, OPS Diagnostics, LLC, Lebanon, NJ, USA) in a 2 mL microtube. The solution of 5% SDC was added to the cell samples for efficient protein extraction using a bead beater (Beadbug microtube homogenizer, Benchmark Scientific, Edison, NJ, USA) at 4 °C. The bead beater was set at 4000 revolutions/min for 30 s, followed by a 30 s pause to cool down. This step was repeated five times. Then, cell lysate was sonicated in an ice-water bath for 30 min to improve the protein dissolution. After that, samples were centrifuged at 21,000 g for 10 min, and the supernatant was collected and diluted twenty times with 50 mM ABC buffer to diminish the interference of 5% SDC with proteomics analysis.

#### 4.7.2. Protein Digestion

Prior to tryptic digestion, the protein concentration of the diluted lysed GBM cell samples was determined by the micro-BCA protein assay, following the manufacturer’s instructions (Thermo Scientific/Pierce, Rockford, IL, USA). A 15 µg aliquot of extracted proteins from each GBM cell sample was subjected to reduction, alkylation, and tryptic digestion. Proteins were first thermally denatured at 80 °C for 30 min. The reduction of proteins was accomplished by adding a 1.25 µL aliquot of 200 mM DTT solution (prepared in 50 mM ammonium bicarbonate (ABC) buffer) and incubating at 60 °C for 45 min. The reduced proteins were alkylated by adding a 5 µL aliquot of the IAA solution (prepared in 50 mM ABC buffer) and incubated at 37.5 °C in the dark for 45 min. A second 1.25 µL aliquot of the DTT solution was added to the samples and incubated at 37.5 °C for 30 min to quench the excessive IAA. Next, a 0.6 µg aliquot of Trypsin/Lys-C Mix was added to the reduced and alkylated proteins (enzyme/substrate ratio of 1:25 *w*/*w*) and incubated at 37.5 °C for 18 h. After incubation, FA (a final concentration of 0.5% *v*/*v*) was added to the samples to quench the enzymatic reaction and precipitate the SDC detergent, which was used in the cell lysis step. Samples were mixed thoroughly, and the supernatant was then collected by centrifugation at 14,800 rpm for 10 min. The supernatants were then speed-vacuum dried and resuspended in 2% ACN and 0.1% FA prior to LC-MS/MS proteomics analysis.

#### 4.7.3. Liquid Chromatography–Mass Spectrometry (LC-MS/MS) Proteomics Analysis

The tryptic digest, corresponding to 1 μg of proteome for each sample, was injected into a 3000 Ultimate nano-LC system (Thermo Fisher Scientific, San Jose, CA, USA) interfaced to an LTQ Orbitrap Velos mass spectrometer (Thermo Fisher Scientific, San Jose, CA, USA) equipped with a nano-ESI source. To remove possible salts, the samples were purified online using a trap column (Acclaim PepMap100 C18 cartridge, 75 µm I.D. × 2 cm, 3 µm particle sizes, 100 Å pore sizes, Thermo Scientific, San Jose, CA, USA). The purified samples were then separated using an Acclaim PepMap100 C18 capillary column (75 µm I.D. × 15 cm, 2 µm particle sizes, 100 Å pore sizes, Thermo Fisher Scientific, San Jose, CA, USA). The column temperature was set to 29.5 °C. Mobile phase A was 2% ACN in water with 0.1% FA, while mobile phase B was 100% ACN with 0.1% FA. To separate peptides, a gradient of 120 min was used at the 350 nL/min flow rate. The gradient of mobile phase B was set as follows: 0–10 min, 5% B; 10–65 min, 5–20% B; 65–90 min, 20–30% B; 90–110 min, 30–50% B; 110–111 min, 50–80% B; 111–115 min, 80% B; 115–116 min, 80–5% B; and 116–120 min, 5% B. The resolution of a full MS was set to 60,000, with the mass-to-charge (*m*/*z*) range set to 400–2000. The data-dependent acquisition mode was employed to achieve two scan events. For the tandem mass spectrometry MS/MS scan, the collision-induced dissociation (CID) was performed on the top 10 most intense ions in a full MS scan event, with a normalized collision energy of 35%, Q-value of 0.25, and activation time of 10 ms. A repeat count of 2, repeat duration of 30 s, exclusion list size of 200, and exclusion duration of 90 s were set for dynamic exclusion.

#### 4.7.4. Protein Identification and Quantification

The LC-MS/MS raw data were analyzed using MaxQuant software (version 2.0.3, Matrix Science Inc., Boston, MA, USA), where the quantitation of proteins was based on ion intensity. The LC-MS/MS data were searched versus the Swiss-Prot human database. Carbamidomethylation of cysteine was set as a fixed modification, while oxidation of methionine and protein N-terminal acetylation were set as variable modifications. Peptides were searched with a precursor mass tolerance of 6 ppm and a fragment mass tolerance of 0.5 Da. The minimal peptide length was set to 7 amino acids, with maximum missed cleavages of two. Only proteins with more than two identified peptides were considered. For the identification of the peptides and proteins, the false discovery rate (FDR) was set to 0.01. Unique peptides and minimal numbers of razors were set to one, and the “matching-between-run” function was authorized. The label-free quantification (LFQ) approach with at least two ratio counts was utilized to compare and normalize the protein intensities across runs.

Perseus version 1.5.5.0 (Max Planck Institute of Biochemistry, Munich, Germany), a supplementary software of MaxQuant, was used to accomplish the statistical analysis. In Perseus, identified proteins from reversed sequences and contaminants were first removed, then those proteins detected in over 70% of the run in at least one sample group remained. After the assessment, the LFQ intensities were log-transformed to simplify the values.

### 4.8. Western Blot Analysis

Proteins were quantified using the Bradford Assay (Bio-Rad). Samples were prepared by adding Laemmli containing *β*-mercaptoethanol, then heat blocked on a thermoshaker at 95 °C for 10 min. Next, 50 μg of the samples was loaded into sodium dodecyl sulfate (SDS)-polyacrylamide gels and then electrophoresed. Membranes were then blocked in 5% non-fat milk prepared in TBS-T (Tris-buffered saline (TBS) and Tween-20) for one hour at room temperature, followed by overnight primary antibody incubation at 4 °C. The antibodies used were rabbit polyclonal anti-POLA1(1:500, Abcam, AB31777, Cambridge, UK), rabbit polyclonal anti-PARP (1:1000, Santa Cruz Biotechnology, SC7150, Dallas, TX, USA), and rabbit polyclonal anti-γH2AX (1:1000, Cell Signaling, CS-2577S, Danvers, MA, USA). Membranes were then incubated with the diluted (1:5000) Horseradish Peroxidase (HRP)-conjugated secondary antibody anti-mouse (Santa Cruz Biotechnology, SC-516102), mouse anti-rabbit (Santa Cruz Biotechnology, SC-2357), or donkey anti-goat (Santa Cruz Biotechnology, SC-2033) for 1 h at room temperature. To assess equal loading, hybridization with rabbit polyclonal beta-actin (1:5000, Abcam, ab8227) and glyceraldehyde 3-phosphate dehydrogenase (GAPDH)-HRP-coupled antibody (1:20,000; Abnova, MAB5476, Taipei, Taiwan) was performed for 30 min at room temperature. Images were taken using a ChemidocTM MP Imaging System (Bio-Rad, Hercules, CA, USA).

### 4.9. Bioinformatic Analysis

#### 4.9.1. Pathway Analysis

For the interactome and pathway analysis, we used Elsevier’s Pathway Studio version 10.001 (https://www.elsevier.com/solutions/pathway-studio-biological-research) accessed on 1 July 2023. To establish the various relationships among the different mass spec-identified proteins relevant to U251-treated and control cells, the ResNet Pathways Studio Propriety database (consisting of relationships between proteins, collated from gene ontology and PubMed literature) was utilized to infer all the interactions. The proteome interactome network was generated using a “direct interaction” algorithm for cellular processes and biological process mapping, as well as the proposed pathway interactions (apoptosis, cell death, cell survival, cell growth and proliferation, cell cycle, cell cycle arrest, malignant transformation, DNA damage, DNA repair, DNA replication, and oxidative stress). For the statistical analysis, the Pathway Studio tool utilizes Fisher’s statistical test to determine if there are nonrandom associations between two categorical variables organized by specific relationships (protein interaction and biological process). The algorithm compares the sub-network distribution to the background distribution using a one-sided Mann–Whitney U test. It calculates a *p*-value indicating the statistical significance of the difference between two distributions. In our analysis, “GenBank ID” and gene symbols from each set were imported to the software to form an experimental data set. For this work, we selected all *p*-values less than 0.005 (Biological Process) and selected overlapping genes greater than 2. The raw data of the pathways selected with the statistical analysis (*p*-value < 0.005) are provided in the Appendix A labeled as (supplementary raw data). The raw data contain the gene ID, references used to build these relations, and the type of interaction (binding, post-translational modification, modulation, inhibition etc.), along with the *p*-value associated with each gene–gene interaction and gene–molecular function interaction.

In addition, we applied MetaScape Software [52] analysis to interrogate the gene overlap among the treatment and controls (Appendix A), to evaluate gene ontology clustering (Appendix A), and, finally, to assess gene interaction–network annotation function (Appendix A).

#### 4.9.2. Principal Component Analysis (PCA)

A Principal Component Analysis (PCA) was employed to gain further insights into the global variations occurring in the treated and control U251, U118, and U87MG cells, using OriginPro 2022b (64-bit) SR1 9.9.5.171 (academic version).

#### 4.9.3. Volcano Plot

The volcano plots were used to visualize the result of differential protein expressions in U251, U87MG, and U118 cells at different treatment conditions versus the control groups, using GraphPad Prism version 9.

#### 4.9.4. Hierarchical Heatmap Clustering

Differences in individual proteins with significant expressions between treated and control U251, U118, and U87MG cells were demonstrated using heatmaps and hierarchical clustering, using Genesis software (version 1.8.1).

#### 4.9.5. In Silico Analysis

To investigate the levels of POLA1 in glioblastoma, we performed in silico analysis using Oncomine, a web-based data-mining platform that allows the user to search several freely available cancer microarray online databases. We selected studies comparing GBM tissues to normal brain counterparts with a *p*-value < 0.05. In silico analysis was conducted using GEPIA (Gene Expression Profiling Interactive Analysis) to evaluate the levels of expression of other genes of interest. This web server offers the RNA sequencing levels of 9736 tumors and 8587 normal samples from The Cancer Genome Atlas (TCGA) and the Genotype-Tissue Expression (GTEx) projects (http://gepia.cancer-pku.cn/ accessed on 2 May 2023). We used The Human Protein Atlas (https://www.proteinatlas.org/ accessed on 2 May 2023) to explore the protein levels in the normal brain (cerebral cortex) compared to glioma tissue immunohistochemistry (IHC) using the tissue atlas (retrieves its data from non-diseased tissue sites) and pathology atlas (data from 17 different human cancers [53]).

### 4.10. Statistical Analysis

Statistical analysis was performed using Microsoft Excel 2010 and GraphPad Prism version 9. Data presented are the means ± SEM of at least three independent experiments. Student’s *t*-test was used to analyze the significance of data for the TUNEL results and select proteins with statistically significant differences between different groups (control versus treated) of each of the three cell lines (U251, U118, and U87MG). In the pathway analysis, Fisher’s statistical test was applied by SNEA to associate one differential hit (validated proteins) to a specific pathway, biological process, or disease. The significance of other experiments was analyzed using two-way ANOVA. Statistical significance was reported when the *p*-value was < 0.05 (* *p* < 0.05; ** *p* < 0.01; *** *p* < 0.001).

## Figures and Tables

**Figure 1 ijms-24-14069-f001:**
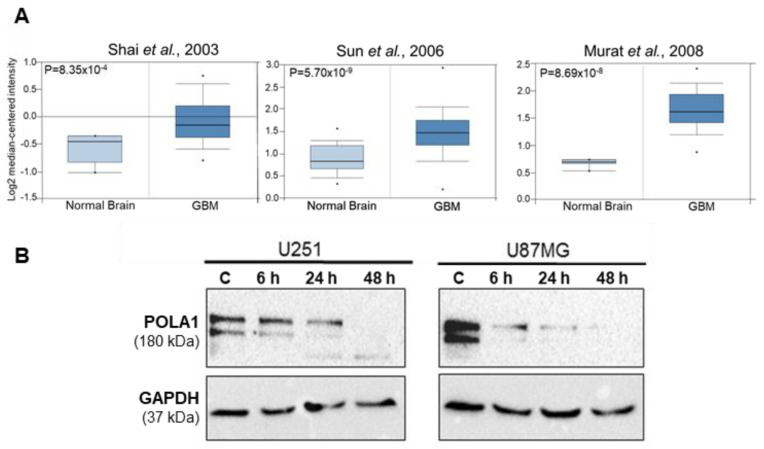
POLA1 levels in GBM tissues and ST1926-treated GBM cell lines. (**A**) In silico analysis identified three studies showing elevated POLA1 expression levels in GBM versus normal brain tissues [14,15,16]. Data generated from oncomine.org; * are outliers. *p* values < 0.05 are considered statistically significant. (**B**) ST1926 reduced POLA1 protein levels in GBM cell lines. U251 and U87MG cells were treated with 0.5 μM ST1926 and immunoblotted against POLA1. Blots were re-probed with GAPDH antibody to ensure equal protein loading (mass spectrometry data are presented in Appendix A). Blots are representative of three independent experiments.

**Figure 2 ijms-24-14069-f002:**
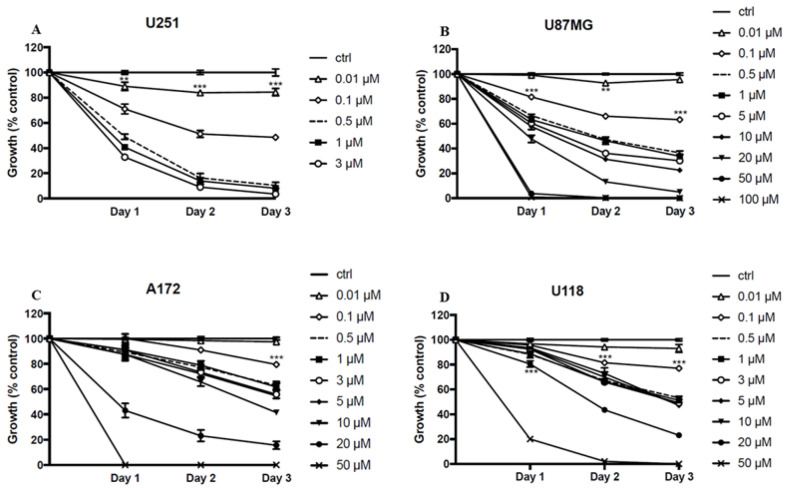
Cell growth inhibition mediated by ST1926 in human GBM cell lines. Cell growth was assessed with the MTT assay. (**A**) U251, (**B**) U87MG, (**C**) A172, and (**D**) U118. Results are represented as the mean ± SEM of three independent experiments, ** *p* < 0.01, *** *p* < 0.001.

**Figure 3 ijms-24-14069-f003:**
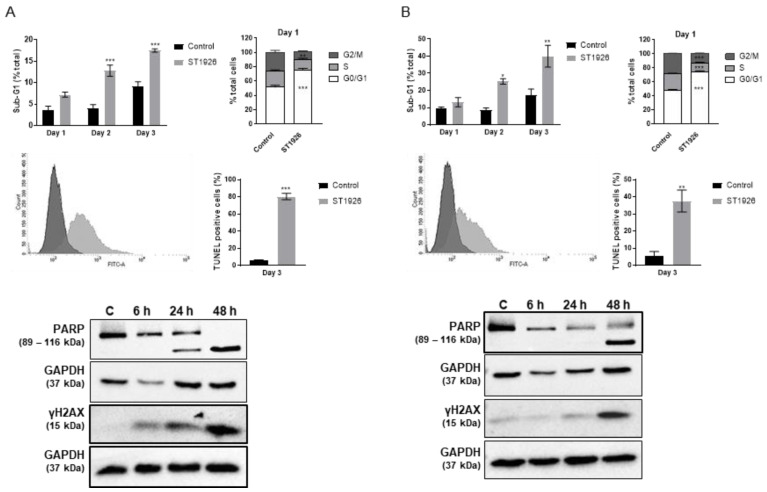
ST1926 treatment of GBM cells induces G_0_/G_1_ arrest, apoptosis, and DNA damage. (**A**) U251 and (**B**) U87MG cells were treated with 0.5 µM ST1926. Cells were stained with propidium iodide and quantified by flow cytometry. Analysis was conducted using BD FACSDiva 8.0. Results are presented as the mean of three independent experiments ± SEM, * *p* < 0.05, ** *p* < 0.01. TUNEL analysis: Representative histogram of the shift from control cells (dark grey) to TUNEL-positive ST1926-treated cells (light grey) of U251 cells. Quantification of TUNEL-positive cells is presented as the mean of three independent experiments ± SEM, *** *p* < 0.001. Whole SDS lysates were immunoblotted against PARP and γ-H2AX antibodies. Blots were re-probed with GAPDH antibody to ensure equal protein loading. Blots are representative of three independent experiments.

**Figure 4 ijms-24-14069-f004:**
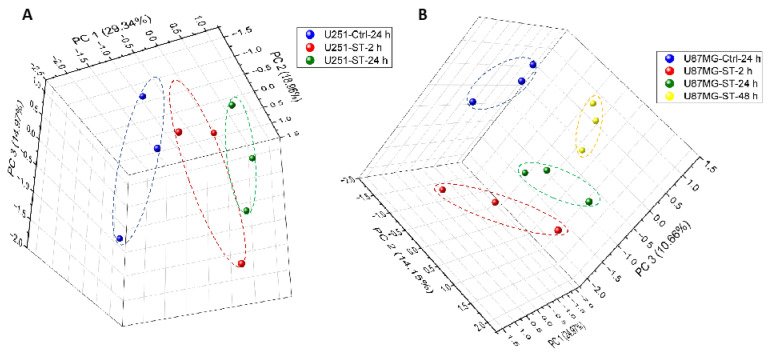
Principal component analysis (PCA) of protein profiles. The PCA plots of protein profiles of (**A**) Control (Ctrl) and U251 cells treated with ST1926 (ST) for 2 and 24 h, and (**B**) Control (Ctrl) and U87GM cells treated with ST1926 (ST) for 2, 24, and 48 h, each obtained from three independent experimental samples.

**Figure 5 ijms-24-14069-f005:**
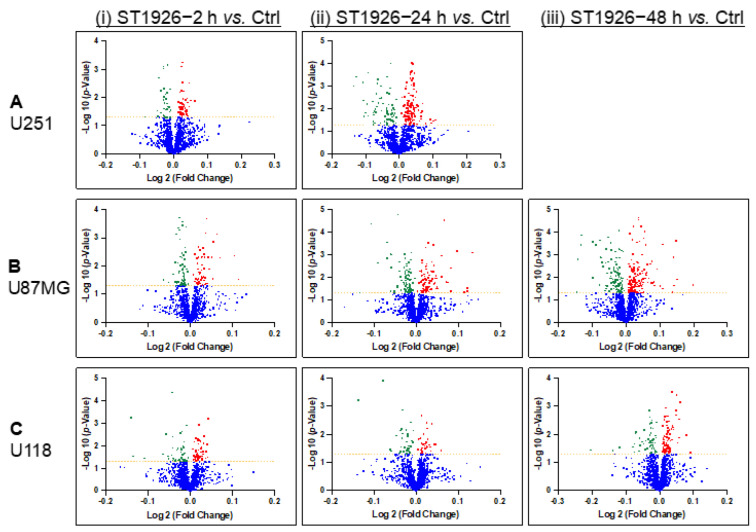
Volcano plots showing the distribution of quantified proteins in (**A**) U251, (**B**) U87MG, and (**C**) U118 cells at (**i**) 2 h, (**ii**) 24 h, and (**iii**) 48 h treatment with ST1926 according to *p*-value and fold change. Significance levels are indicated with yellow lines (at *p*-value < 0.05) and color-coded dots (red represents the up-regulated proteins, and green represents the down-regulated proteins, and blue represents the proteins without significant expression).

**Figure 6 ijms-24-14069-f006:**
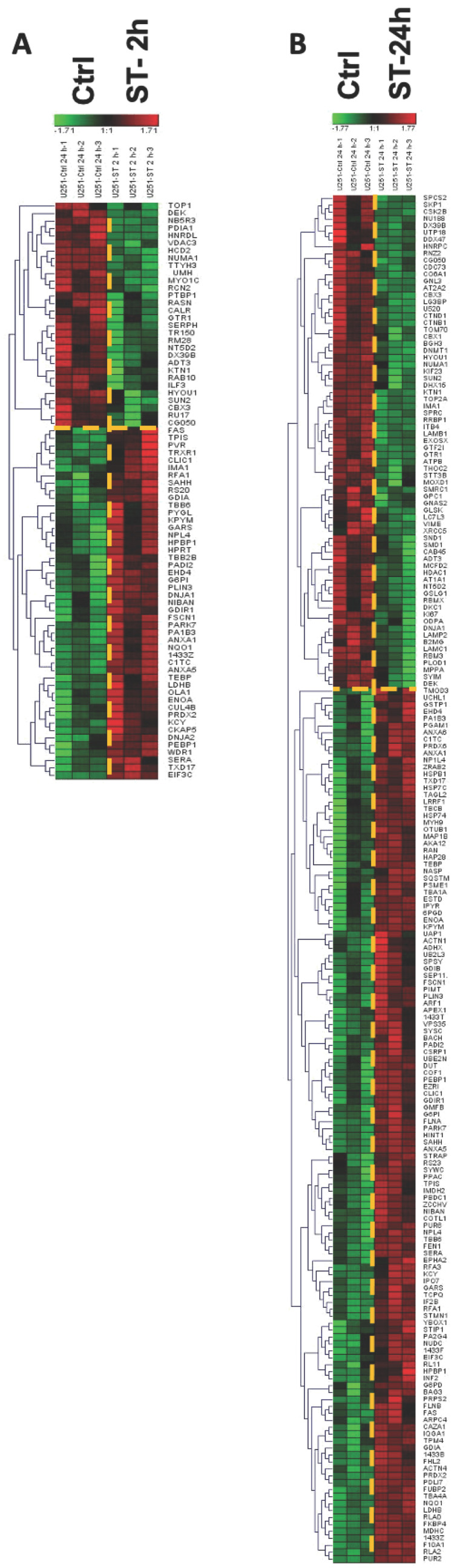
Heatmaps of proteins whose expression was significantly altered upon treating U251 cells with ST1926 (ST) at (**A**) 2 h and (**B**) 24 h. Each row represents a different protein, while each column represents one of the triplicates of its corresponding experimental group. The expression level is depicted by color and intensity: green indicates down-regulation, red indicates up-regulation, and higher intensity reflects a greater fold change.

**Figure 7 ijms-24-14069-f007:**
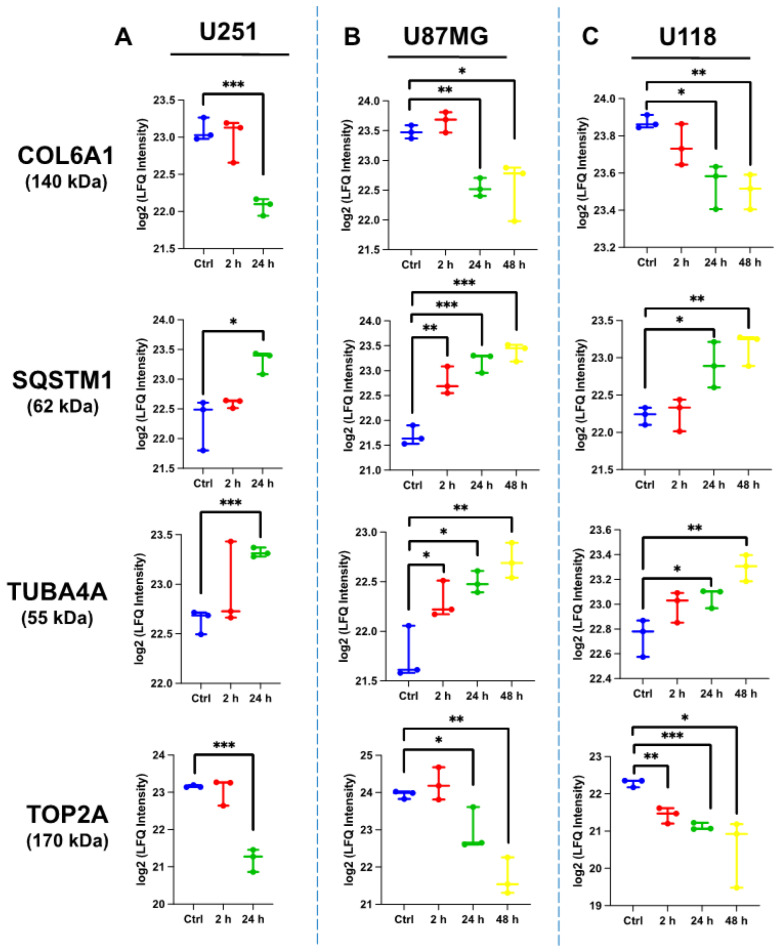
Comparative protein analysis upon ST1926 treatment of the GBM cell lines using mass spectrometry. (**A**) U251, (**B**) U87MG, and (**C**) U118 cells were treated with 0.5 μM ST1926 at the indicated time points. *, *p* < 0.05; **, *p* < 0.01; *** *p* < 0.001.

**Figure 8 ijms-24-14069-f008:**
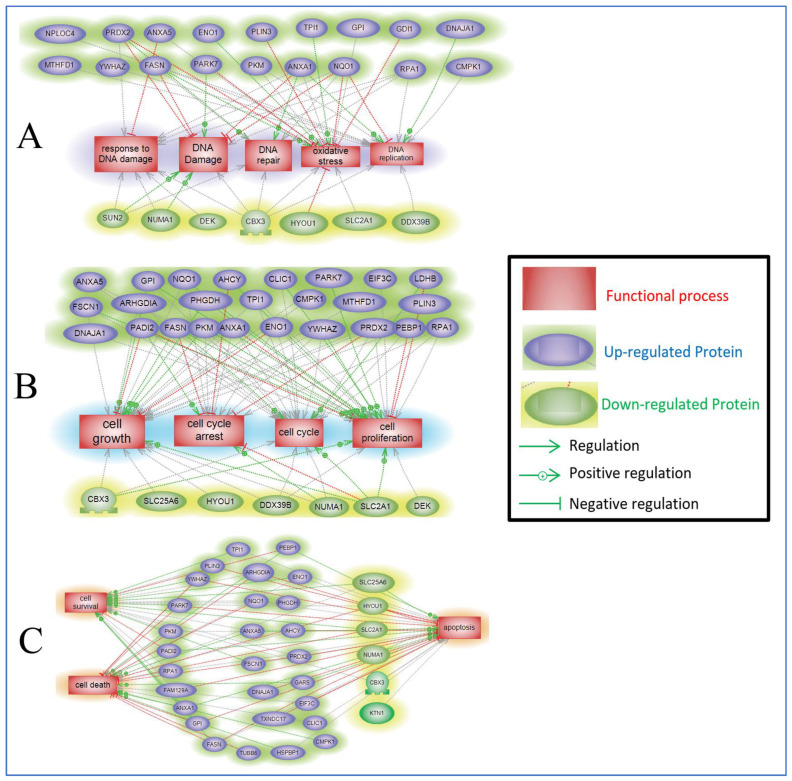
Pathway Studio Analysis of the differentially expressed proteins between control and U251 cells treated with ST1926. Proteins in blue color are up-regulated, while those in green color are down-regulated, and their relationships with (**A**) DNA replication, DNA repair, DNA damage, response to DNA damage, and oxidative stress; (**B**) cell proliferation, cell growth, cell cycle, and cell cycle arrest; and (**C**) cell survival, apoptosis, and cell death are indicated. The raw data with the *p-*values of the relationship and pathways indicated in each of the generated panels are provided as supplementary raw data.

**Figure 9 ijms-24-14069-f009:**
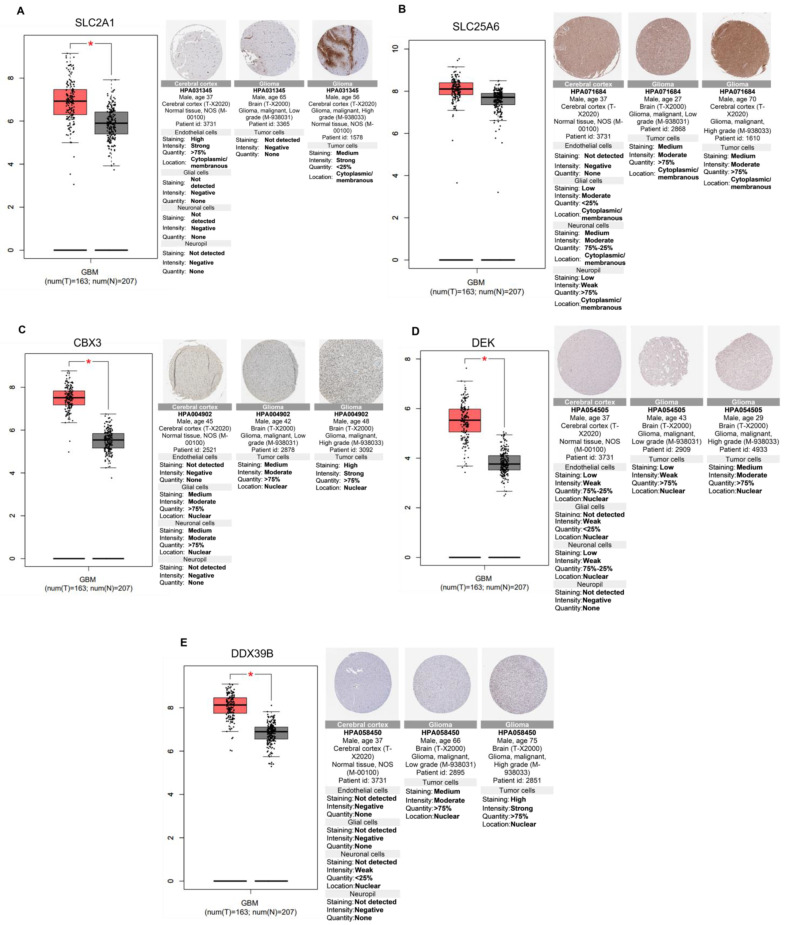
Elevated gene and protein expression between normal brain and GBM tumor tissues. The box plots represent the transcripts level of the five different genes in glioblastoma tissue (red box) versus normal brain tissues (grey box), based on the Gene Expression Profiling Interactive Analysis (GEPIA) database, with tumor and normal tissue samples from the Cancer Genome Atlas (TCGA) and Genotype-Tissue Expression (GTEx) portals. The number of tumors (T) and normal (N) tissues is indicated; log2 (TPM+1) was used for log-scale, * *p*-value < 0.05, one-way ANOVA. Using The Human Protein Atlas, the immunohistochemical comparison of the same genes is represented next to the box plots in both normal brains (cerebral cortex) and glioma tissues. Gene and protein levels are presented of (**A**) Solute carrier family 2, facilitated glucose transporter member 1 (SLC2A1); (**B**) Solute carrier family 25 members (SLC25A6); (**C**) Chromobox protein homolog 3 (CBX3); (**D**) DEK oncogene; and (**E**) Spliceosome RNA helicase (DDX39B).

**Figure 10 ijms-24-14069-f010:**
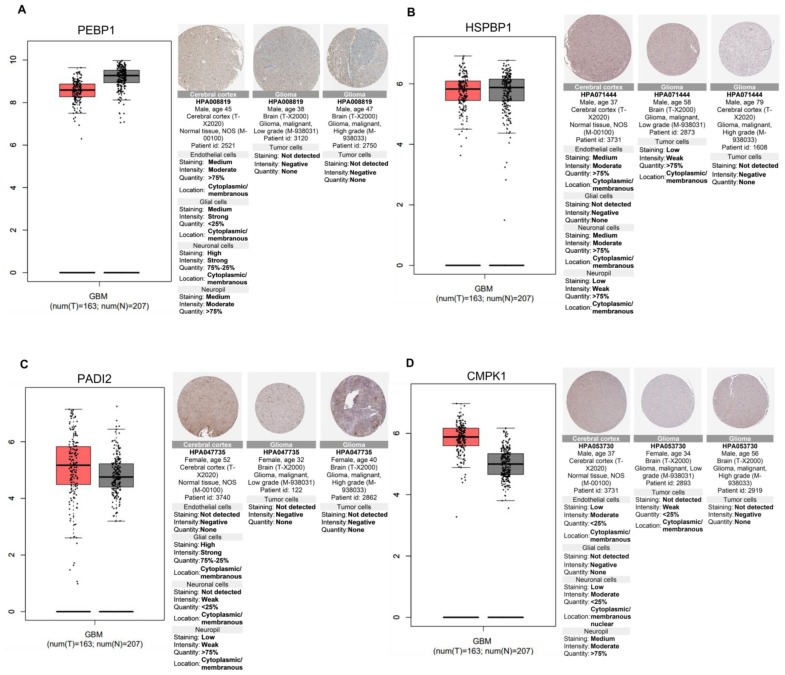
Differential gene and protein expression between normal brain and GBM tumor tissues. The box plots represent the transcripts level of the four different genes in glioblastoma tissue (red box) versus normal tissue (grey box), based on the Gene Expression Profiling Interactive Analysis (GEPIA) database, with tumor and normal tissue samples from the Cancer Genome Atlas (TCGA) and Genotype-Tissue Expression (GTEx) portals. The number of tumors (T) and normal (N) tissues is indicated; log2 (TPM+1) was used for log-scale. Using The Human Protein Atlas, the immunohistochemical comparison of the same genes is represented next to the boxplot in both the normal brain (cerebral cortex) and glioma tissues. Gene and protein levels are presented of (**A**) Phosphatidylethanolamine-binding protein 1 (PEBP1); (**B**) Hsp70-binding protein 1 (HSPBP1); (**C**) Protein-arginine deiminase type-2 (PADI2); and (**D**) UMP-CMP kinase (CMPK1).

**Table 1 ijms-24-14069-t001:** Variation of the number of up-regulated and down-regulated proteins in control (Ctrl) versus ST1926-treated cell lines at the indicated time points (Appendix A).

Cell Line	Conditions	Number of Up-Regulated Proteins	Number of Down-Regulated Proteins
U251	Ctrl vs. 2 h	47	30
	Ctrl vs. 24 h	126	71
U118	Ctrl vs. 2 h	51	45
	Ctrl vs. 24 h	36	35
	Ctrl vs. 48 h	83	53
U87MG	Ctrl vs. 2 h	54	61
	Ctrl vs. 24 h	88	79
	Ctrl vs. 48 h	156	126

**Table 2 ijms-24-14069-t002:** List of proteins according to their appearance in the mass spectrometry analysis of the U251, U87MG, and U118 GBM cell lines after 24 h of treatment with 0.5 μM ST1926.

Common gene alteration among U251, U87MG, and U118 cells (4)
DNA topoisomerase 2-alpha (TOP2A) Sequestosome-1/p62 (SQSTM1) Tubulin alpha-4A chain (TUBA4A) Collagen alpha-1(VI) chain (COL6A1)
Common gene alteration between U251 and U87MG cells (39)
Kinectin (KTN1) Ribosome-binding protein 1(RRBP1) Importin subunit alpha-1 (KPNA) General transcription factor II-I (GTF2I) Proliferation marker protein Ki-67 (MKI67) Galectin-3-binding protein (LGALS3BP) DnaJ homolog subfamily A member (DNAJA1) Golgi apparatus protein 1(GLG1) Procollagen-lysine,2-oxoglutarate 5-dioxygenase 1 (PLOD1) S-phase kinase-associated protein 1 (SKP1) Laminin B2 chain (LAMB2) Staphylococcal nuclease domain-containing protein 1 (SND1) Nuclear mitotic apparatus protein 1 (NUMA1) Guanine nucleotide-binding protein G(s) subunit alpha isoforms short (GNAS) Chromobox protein homolog 1 (CBX1) Solute carrier family 25 member (SLC25A6) X-ray repair cross-complementing protein 5 (XRCC5) 60S ribosomal protein L11 (RPL11) Tryptophan-tRNA ligase (WARS1) 40S ribosomal protein S23 (RPS23) Y-box-binding protein 1 (YBX1) A-kinase anchor protein 12 (AKAP12) Spermine synthase (SMS) Heat shock cognate 71 kDa protein (HSPA8) Fatty acid synthase (FAS) Proliferation-associated protein 2G4 (PA2G4) Triosephosphate isomerase (TIM) Eukaryotic translation initiation factor 3 subunit C (EIF3C) Inosine-5′-monophosphate dehydrogenase 2 (IMPD 2) 14-3-3 protein beta/alpha (YWHAB) Perilipin-3 (PLIN3) Annexin A1 (ANXA1) ADP-ribosylation factor 1 (ARF1) Stathmin (STMN1) FUSE-binding protein 2 (FUBP2) 60S acidic ribosomal protein P2 (RPLP2) Hsc70-interacting protein (FAM10A1) Filamin-A (FLNA) Trifunctional purine biosynthetic protein adenosine-3 (GART)
Common gene alteration between U251 and U118 cells (6)
Lysosome-associated membrane glycoprotein 2 (LAMP2) SUN domain-containing protein 2 (SUN2) Spliceosome RNA helicase (DDX39B) Nucleosome assembly protein 1-like 4 (NAP1L4) Histidine triad nucleotide-binding protein 1 (HINT1) Adenosylhomocysteinase (AHCY)
Common gene alteration between U87MG and U118 cells (10)
Endoplasmin (HSP90B1) Guanine nucleotide-binding protein G(i) subunit alpha-2 (GNAI2) Signal transducer and activator of transcription 3 (STAT3) X-ray repair cross-complementing protein 6 (XRCC6) Exportin-1 (XPO1) Transportin-3 (TNPO3) Catenin alpha-1 (CTNAA1) 4F2 cell-surface antigen heavy chain (SLC3A2) Vesicle-associated membrane protein-associated protein A (VAPA) Elongation factor 1-gamma (EF1G)

## Data Availability

Not applicable.

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
