# Peer review of "The Antitumor Effect of the DNA Polymerase Alpha Inhibitor ST1926 in Glioblastoma: A Proteomics Approach"

_ijms, 2023, doi:10.3390/ijms241814069_

Round 1

Reviewer 1 Report

The manuscript by El-Baba et al. reports on the effects of the synthetic ST1926 retinoid in glioblastoma using a number of cell-lines.  In addition, the manuscript provides information regarding the actin of ST1926 on the levels of various proteins, using a high-throughput proteomic approach.  Overall the manuscript is well-written and the data are presented in a clear manner.  No further experiments are required.  However, the manuscript should be carefully edited for english language, as there are a number of mistakes and unclear statements in the Abstract, Introduction and Results sections.  For instance, in the abstract,  lines 25-26 should be rephrased. 

The manuscript should be carefully edited for english language, as there are a number of mistakes and unclear statements in the Abstract, Introduction and Results sections.

Author Response

The manuscript by El-Baba et al. reports on the effects of the synthetic ST1926 retinoid in glioblastoma using a number of cell-lines. In addition, the manuscript provides information regarding the actin of ST1926 on the levels of various proteins, using a high-throughput proteomic approach. Overall the manuscript is well-written and the data are presented in a clear manner. No further experiments are required.  However, the manuscript should be carefully edited for english language, as there are a number of mistakes and unclear statements in the Abstract, Introduction and Results sections.  For instance, in the abstract,  lines 25-26 should be rephrased.

Comments on the Quality of English Language

The manuscript should be carefully edited for english language, as there are a number of mistakes and unclear statements in the Abstract, Introduction and Results sections.

The manuscript has been revised by the English Editor throughout the entire document.

Reviewer 2 Report

Dear Authors and Editors. 

The paper is focused on identification of mechanisms of actions for ST1926 agent against glioblastoma using several cell lines. 

Previous publications of the authors reported the effect of ST1926 on mitochondria functions and demonstrated that ST1926 is a DNA polymerase 2 inhibitor. 

The current work reports effects on cell viability and cell cycle, decrease in polymerase 2 levels and performs a proteomics analysis of the data. 

Below are several points that must be addressed before the publication can be considered for publication. 

The strong part of the investigation is the demonstration that the levels of POL2 are going down. 

  1. The western demonstrating equal loading by  GAPDH for U87MG leaves a lot of room for improvement…, need to repeat the experiment and add Ponceau S staining to demonstrate equal loading also taking into account that GAPDH does not change as suggested by the Supplementary figure.

  2. The figure 2 is good, although, needless to say, MTT reflects mitochondrial activity rather than cell growth.

  3. The effects on cell cycle and are also not bad (Figure3), although the cell cycle figures merging S and G2/M phases need to be supported by conventional distributions of DNA quantities and G2/M and S phases must be separated. Please perform BrDu or similar staining of cells in the S phase. 

  4. Figure 4 is designed to troll a reviewer and not informative. How can the reader interpret this? Shall he look up every protein id in this??? 

  5. Table 2. The reader has to infer if proteins go up or down right?

  6. Figure 7 - I would suggest trying the Metascape software, supplying both up and downregulated proteins in a single .txt file in two columns. This will help visualization of protein interaction clusters and annotations.

I have an impression that authors struggle to make a conclusion from all this work.  "These identified new targets may become potential biomarkers in future strategies for treating GBM". Does it have the real ground behind? 

What does the fact that some proteins affected by ST1926 in glioblastoma cell lines are also misregulated in tumors teaches us?

I think the authors need to re- evaluate their results and spend much more time and effort shaping the paper themselves instead of leaving this work for reviewers.

Sincerely

Author Response

The paper is focused on identification of mechanisms of actions for ST1926 agent against glioblastoma using several cell lines.

Previous publications of the authors reported the effect of ST1926 on mitochondria functions and demonstrated that ST1926 is a DNA polymerase 2 inhibitor.

The current work reports effects on cell viability and cell cycle, decrease in polymerase 2 levels and performs a proteomics analysis of the data.

Below are several points that must be addressed before the publication can be considered for publication.

The strong part of the investigation is the demonstration that the levels of POL2 are going down.

We thank the reviewer for these comments and suggestions that will definitely improve the quality of our manuscript. We have answered point by point all the revisions as per below.

The western demonstrating equal loading by GAPDH for U87MG leaves a lot of room for improvement…, need to repeat the experiment and add Ponceau S staining to demonstrate equal loading also taking into account that GAPDH does not change as suggested by the Supplementary figure.

We have repeated the experiment and presented a new western blot, as shown in the new Figure 1B. The full blot and the corresponding Ponceau staining are included in the Revised Full Blots, clearly showing equal protein loading.

The figure 2 is good, although, needless to say, MTT reflects mitochondrial activity rather than cell growth.

 We agree with the reviewer that MTT reflects mitochondrial activity, which is an indirect indicator of cell growth. We have clarified this point in the Figure 2 legend.

We have validated the MTT results in two of the selected cell lines using the sulforhodamine B (SRB) assay for cell density determination based on the measurement of cellular protein content (an indirect measure of cell growth). We observed similar trends upon ST1926 treatment. These results are presented in Figure S2.

The effects on cell cycle and are also not bad (Figure3), although the cell cycle figures merging S and G2/M phases need to be supported by conventional distributions of DNA quantities and G2/M and S phases must be separated. Please perform BrDu or similar staining of cells in the S phase.

As suggested by the reviewer, we have separated the S and G2/M phases as shown in the new Figure 3A and 3B. The cell cycle analysis results mostly indicate a G0/G1 cell cycle arrest in U251 and U87MG cell lines treated with ST1926. Only U87MG-treated cells displayed a significant S phase arrest.

We started performing BrDu staining of cells, but we ran out of reagents, and we were not able to finalize the experiments. Unfortunately, it takes a long time to ship the reagents to Lebanon. We mentioned in the text that BrDu staining would confirm the effect of ST1926 treatment on the S-phase distribution of the cells.

Figure 4 is designed to troll a reviewer and not informative. How can the reader interpret this? Shall he look up every protein id in this???

Thank you for your comments. Hierarchical heatmap clusterings are now provided using Gene IDs instead of Protein IDs to help the reader interpret the data more efficiently. Figure 4 is changed to Figure 6 in the current version of the manuscript. Therefore, please check Figure 6 and also Figures S4 and S5 in the Supplementary document.

We have also updated the Systems Biology analysis of the interaction protein molecular function map. We apologize that we did not include the raw data in our previous submission. An annotation of each relationship between the altered differential gene/protein is now designated with the corresponding relation supported with the reference from literature along with the p-value associated with each protein-protein interaction and protein-molecular function interaction (These are shown in the supplementary raw data). This is the main feature of the Elsevier Software Pathway Studio; it indicates and explains each of the projected relations.

In addition, we have modified the figures with enhanced coloring that indicates the up-regulation and down-regulation, where the blue color indicates up-regulation and green indicates down-regulation. It is of note that the interactive map and molecular function figure is changed to Figure 8.

Table 2. The reader has to infer if proteins go up or down right?

 The lists of up- and down-regulated proteins in each comparison are provided in Tables S1-S8, along with their accession numbers, protein names, gene IDs, p-values, and fold changes. Please check them in the Supplementary documents.

Similarly, the Figures are updated with better visualization that shows clearly how these proteins are up-regulated or being down-regulated. Additionally, to elaborate on how these relations are inferred, we have supplemented the new submission with the raw data (These are shown in the supplementary raw data). This supplemental data contains the gene IDs, References used to build these relations, and the type of interaction (binding, post-translational modification, modulation, inhibition, etc.). Annotation of each relationship between the altered differential protein is now designated with the corresponding relation supported with the reference from literature along with the p-value associated with each protein-protein interaction and gene-molecular function interaction.

Figure 7 - I would suggest trying the Metascape software, supplying both up and downregulated proteins in a single .txt file in two columns. This will help visualization of protein interaction clusters and annotations.

 (Please note that Figure 7 has been changed to Figure 8 in the current version of the manuscript)

We thank the reviewer for this constructive suggestion. We are aware of the NIH-Funded Metascape software and its excellent utility in demonstrating relationships among proteins via (1) the Circos Plot that can indicate how proteins from the input ST1926 versus the control overlaps, which is now is added as a new Figure S7. In addition, we have performed Gene Ontology Clustering input (Figure S8) ST1926 versus the control, where we found that the top hit enriched pathway is the Cell cycle pathway, RNA metabolism, and other signaling pathways that overlap with the targetted analysis we performed with Pathway Studio (Figure 8). Finally, we applied the protein-protein interaction-network annotation function (Figure S9 ), where MetaScape will be using the "Molecular Complex Detection" (MCODE) algorithm to generate these networks. Again, the top network identified involved RNA interactions/assembly and metabolic reprogramming in cancer. These new supplementary figures are added to the manuscript.

We have also added a reference for the use of Metascape software:

Zhou et al. Metascape provides a biologist-oriented resource for the analysis of systems-level datasets. Nature Communications 2019.

One important feature of Pathway Studio software is that it provides raw data on how these pathways and interaction maps are generated and how these relations are inferred. The raw data contain the gene IDs, References used to build these relations, and the type of interaction (Binding, post-translational modification, modulation, inhibition, etc.) along with the p-value associated with each protein-protein interaction and protein-molecular function interaction.

This particular feature makes us so confident to use this bioinformatic analysis. We have published more than 20 papers analyzing Omics data using the same approach. These include genomics and proteomics data as illustrated below.

Published work from our team using Pathway Studio:

  1. Sylvain Osien, Alice Capuz,. Marie Duhamel, David Devos, Firas Kobeissy, Fabien Van den Abelle, Amélie Bonnefond, Isabelle Fournier, Franck Rodet, Michel Salzet Heimdall, an alternative proteiorganon issued from a ncRNA related to kappa light chain variable region of immunoglobulins from astrocytes: a new player in neural proteome" Cell Death & Disease 2023 Aug 16;14(8):526.
  2. Alice Capuz, Sylvain Osien, Melodie karnoub, Soulaimane Aboulouard, Estelle Laurent, Etienne Coyaud, Antonella Raffo Romero, Firas Kobaissy, Fabien Vanden Abeele, Isabelle Fournier, Dasa Cizkova, Franck Rodet, and Michel Salzet. Astrocytes express aberrant immunoglobulins as putative gatekeeper of astrocytes to neuronal progenitor conversion; Cell Death & Disease; 2023 Apr 4;14(4):237.
  3. Khalil Mallah, Kazem Zibara, Coline Kerbaj, Ali Eid, Nour Khoshman, Zahraa Ousseily, Abir Kobeissy, Tristan Cardon, Dasa Cizkova, Firas Kobeissy, Isabelle Fournier, Michel Salzet Neurotrauma Investigation through spatial omics guide by mass spectrometry imaging: Targets identification and clinical applications Mass Spectrom Rev. 2023 Jan;42(1):189-205. 577616.
  4. Malaker SA, Quanico J, Raffo-Romero A, Kobeissy F, Aboulouard S, Tierny D, Bertozzi CR, Fournier I, Salzet M, On-tissue spatially resolved glycoproteomics guided by N-glycan imaging reveal global dysregulation of canine glioma glycoproteomic landscape. Cell Chemical Biology 2022, Jan 20;(1): 30-42.
  5. Mélanie Rose, Tristan Cardon, Soulaimane Aboulouard, Nawale Hajjaji, Firas H Kobeissy, Marie Duhamel, Isabelle Fournier, Michel Salzet Surfaceome proteomic of glioblastoma revealed potential targets for immunotherapy, Front. Immunol. 2021, Sep 27;12:746168.
  6. Aboulouard S, Wisztorski M, Duhamel M, Saudemont P, Cardon T, Narducci F, Lemaire AS, Kobeissy F, Leblanc E, Fournier I, Salzet In-depth proteomics analysis of sentinel lymph nodes from individuals with endometrial cancer.Cell Rep Med. 2021 Jun 15;2(6):100318.
  7. Rouba Hage-Sleiman, Hisham Bahmad, Hadile Kobeissy, Firas Kobeissy, and Ghassan Dbaibo. Genomic alterations in p53-dependent apoptosis induced by γ-irradiation of Molt-4 leukemia cells, PlosOne; 2017 Dec 22;12(12):e0190221.
  8. Spatial analysis of the glioblastoma proteome reveals specific molecular signatures and markers of survival Nat Commun2022 Nov 4;13(1):6665.
  9. Hisham F. Bahmad, Wenjing Peng, Rui Zhu, Farah Ballout, Alissar Monzer, Mohamad K. Elajami, Firas Kobeissy, Wassim Abou-Kheir, and Yehia Mechref; Protein Expression Analysis of an In Vitro Murine Model of Prostate Cancer Progression: Towards Identification of High-Potential Therapeutic Targets J Personalized Medicine 2020; 10;10(3):83.

I have an impression that authors struggle to make a conclusion from all this work.  "These identified new targets may become potential biomarkers in future strategies for treating GBM". Does it have the real ground behind?

Our research is just the beginning. Extensive future functional and mechanistic studies are still needed to determine whether these identified new targets will become potential biomarkers for treating GBM using POLA1 inhibitors. We have omitted the sentence between the quotation from the abstract.

What does the fact that some proteins affected by ST1926 in glioblastoma cell lines are also misregulated in tumors teaches us?

We thank the reviewer for this insightful question. The evidence that some proteins affected by ST1926 are also misregulated in tumors teaches us crucial facts about GBM biology and therapeutic strategies. These identified novel proteins may be potential therapeutic targets for new drug development in GBM, especially using POLA1 inhibitors. We have added the latter sentence to the summary.

I think the authors need to re- evaluate their results and spend much more time and effort shaping the paper themselves instead of leaving this work for reviewers.

We apologize to the reviewer regarding the long time spent reading and evaluating our manuscript. We wholeheartedly thank the reviewer for the comments and insightful suggestions that have enhanced the quality of our manuscript.

Round 2

Reviewer 2 Report

Dear Authors,

Thank you very much for addressing my concerns.